# Historical Evolution of Snowpack Capacity to Buffer Rain-on-Snow Runoff in a Large Columbia River Headwaters Basin

Joel Brown<sup>1,2</sup>, Joel Harper<sup>1</sup>

Department of Geosciences, University of Montana, Missoula, 59812, USA

<sup>2</sup>Aesir Consulting LLC, Missoula, 59801, USA

Correspondence to: Joel Brown (joel@aesirmt.com)

Abstract. Rainfall during the snow season plays an increasingly important role in flood risk as climate warms and extreme events become more frequent. However, a given sized rain-on-snow (ROS) event can yield outcomes ranging from flooding to no runoff, depending partly on the snowpack's antecedent cold content and capillary retention forces. Here, we analyze the seasonal evolution of the snowpack's physical state over a 72-year period to assess long-term changes in its capacity to buffer runoff from liquid water input. We use ERA-5 Land data to force a snowpack model that tracks the layer-by-layer development of heat, mass, and structural framework of the snowpack throughout the snow season. We test our approach in a large Columbia River headwaters basin in NW Montana, USA. We evaluate cold content and total capillary retention of the snowpack to determine long term trends in Liquid Water Buffering Capacity (LW<sub>bc</sub>) as it evolves throughout the snow season. The LW<sub>bc</sub> of the snowpack exhibited robust long-term declines across all elevation bands, despite high intra- and interannual variability. The largest declines occurred during the Spring period, trending downward across the historical period by 43% to 80% depending on the elevation band. The core five weeks of mid-winter showed no trending change of LW<sub>bc</sub>, and in fact demonstrated an increase in cold content over the 72 years. Our findings demonstrate that changes in the snowpack's ability to buffer runoff, including dependencies on local basin factors related to snowpack seasonality and elevation, are a key component of evolving ROS risk.

## 1 Introduction

Many extreme river discharges observed in mountain regions are associated with Rain-On-Snow (ROS) events due to the compounding of runoff from rainfall with snowmelt (Berghuijs et al., 2016; Marks et al., 1998; McCabe et al., 2007; Sui and Koehler, 2001). Consequently, ROS events can play a key role in assessments of flood risk (Li et al., 2019; Musselman et al., 2018) and infrastructure management (Judi et al., 2018; Yan et al., 2018). As climate warms and extreme weather events become more frequent (e.g. Fischer and Knutti, 2016; Robinson et al., 2021), there is increasing attention on the frequency and intensity of large rainfall events occurring during the snow season (Freudiger et al., 2014; Musselman et al., 2018). Numerous studies have therefore addressed how atmospheric conditions driving frequency and intensity of ROS events have evolved in a changing climate (Cohen et al., 2015; Ohba and Kawase, 2020; Pall et al., 2019). Other studies (Haleakala et al., 2023; Marks et al., 1998; Musselman et al., 2017, 2018; Wayand et al., 2015) have directly acknowledged the importance of antecedent conditions of the landscape, streamflow, and snowpack on the effect of ROS flood risk. For instance, a ROS event occurring in Spring, when

melting snow already contributes to widespread runoff across the landscape, may generate markedly different hydrological responses compared to an event occurring in mid-winter.

The physical state of the snowpack at the onset of rain can also cause the runoff generated by a given ROS event to vary significantly. Before a 1D profile of the snowpack reaches a state in which it can release water from its base, two key conditions must first be met: the snow must be warmed to 0 °C, eliminating its cold content, and its pore space must be saturated with enough liquid water to overcome internal capillary forces (Colbeck, 1972). Cold content is primarily removed by the latent heat released during the refreezing of rainwater or infiltrated snow melt water (Colbeck, 1972). The degree of saturation needed to overcome capillary forces depends on the evolving snow microstructure but is typically estimated at 5–7% by volume or 5–15% by weight (e.g. Coléou and Lesaffre, 1998). The buffering effects of both cold content and capillary retention can be quantified as liquid water equivalents, representing the volume of water required to meet the energy and saturation thresholds for flow initiation (DeWalle and Rango, 2008). Here, we term this quantity the Liquid Water Buffering Capacity (LW<sub>bc</sub>). Robust assessments of long-term changes in ROS flood risk must consider not only atmospheric forcing, antecedent land surface and stream conditions, but also how this snowpack LW<sub>bc</sub> has evolved under changing climate conditions.

As near-surface air temperatures rise, most indicators of snowpack stability are in decline (Barnett et al., 2005), though the nature and magnitude of these changes vary by region (Luce et al., 2014; Mote et al., 2005, 2018; Musselman et al., 2021) and elevation (Aguado et al., 1992). Observed changes include earlier peak spring runoff (Barnett et al., 2005), reduced snow persistence (Evan, 2019), declining April 1 Snow Water Equivalent (SWE) (Mote et al., 2005, 2018), and decreasing snow cover extent (Déry and Brown, 2007). However, changes in physical parameters related to the snowpack's LW<sub>bc</sub> are poorly documented, as they are not easily observable at the basin scale or remotely sensed. Moreover, the evolution of LW<sub>bc</sub> throughout the snow season, and long-term changes therein, is vital since ROS events do not necessarily have equal probability of occurring on each day with snow on the ground. Consequently, the lack of information on cold content and capillary resistance complicates efforts to reconcile climatic shifts in ROS runoff characteristics with critical antecedent snowpack conditions.

Evaluating present and future snowpack change requires understanding how the system has evolved over time. This study investigates historical changes in the snowpack's ability to buffer runoff from ROS events. Because essential basin-wide observational data such as daily measurements of cold content do not exist, we utilize a 72-year model simulation of the seasonal evolution of internal snowpack conditions. Modeling LWbc parameters is computationally expensive, requiring layer-by-layer tracking and accounting for high temporal and spatial variability, including microstructural evolution. Broad regional-scale studies are not only computationally impractical but may obscure the importance of transient and localized processes. We therefore conduct detailed modeling and analysis of long-term trends in snowpack cold content and capillary retention in a large Columbia River headwaters basin in northwest Montana, USA as a proof-of-concept study. Understanding how snowpack buffering capacity has changed under a warming climate provides critical context for interpreting recent and future hydrologic risks.

# 2 Methods

# 75 2.1 Study basin

85

Our work is focused on the South Fork of the Flathead (SFF) River watershed basin which has an area of approximately 4,340 km<sup>2</sup> and spans an elevation range from 922 m to 2853 m above sea level (Fig. 1). The basin consists of undeveloped forest and alpine areas with a dam-controlled reservoir covering about one fourth of the valley floor. Within the basin, there are four permanent automated weather stations, one of which is a Snow Telemetry (SNOTEL) site located at a relatively low elevation of 1326 m elevation (Emery Creek SNOTEL). Another SNOTEL site is located approximately 500 meters west of the basin ridge line at 1841 m elevation close to the northern end of the basin (Noisy Basin SNOTEL). We include a ~29 km<sup>2</sup> area around this SNOTEL site in our watershed model domain in order to have two sites with different elevations to compare our modeled SWE results against.

Figure 1: Map of the South Fork of the Flathead Basin with modeled area (cross hatched area), ERA5-Land pixel centroids (red triangles), and SNOTEL sites (yellow stars). This map was made using QGIS, USGS 180m DEM, and Montana Fish Wildlife and Parks stream and lake data.

## 2.2 Snowpack simulations

We use Alpine3D (Lehning et al., 2006) to model snowpack evolution physics over the SFF basin with an 800 by 800 m resolution, with this resolution the model elevation range is 944 m to 2594 m above sea level. Alpine3D is a spatially distributed snow allocation and surface hydrology developed by the WSL Institute for Snow and Avalanche Research. The model incorporates modules for resampling/filtering meteorological data (MetIO) (Bavay and Egger, 2014) and a physics based, finite element 1D snowpack evolution model (SNOWPACK) (Lehning et al., 1999) along with internal models for snow transport (SnowDrift), energy balance (EBalance), and runoff. Model forcing parameters listed in Appendix Table A1 are derived from ERA5-Land reanalysis products (Muñoz-Sabater et al., 2021). ERA5-Land is a global reanalysis product based on the land surface component of the fifth generation European ReAnalysis (ERA5) meteorological model provided by the European Centre for Medium-Range Weather Forecasts (ECMWF) which is available globally on a 0.1-degree grid with hourly time steps from 1950 to 2-3 months prior to the present date. Comparison of modeled max SWE and date of max SWE vs. SNOTEL measurements show reasonable agreement when ERA5-Land precipitation input is multiplied by a regional factor of 1.5 (Fig. A1). This regional bias correction of precipitation is based on a simple regression of ERA5-Land and SNOTEL precipitation (Pan et al., 2003) which accounts for well-known uncertainty in land surface model precipitation values, especially in mountainous terrain (e.g. Cho et al., 2022; Pan et al., 2003; Raleigh et al., 2015).

## 2.3 Cold content and LWbc from model outputs

As part of the physics-based model structure of Alpine3D, cold content and SWE of the full snowpack depth are calculated for each time step in the model. Within the model, cold content (CC) is calculated for each layer by Eq. (1):  $CC = c_i \rho_s d_s (T_s - T_m)$ , (1)

where  $c_i$  is the specific heat of ice,  $\rho_s$  is the snow density,  $d_s$  is depth of snow, and  $T_s$  and  $T_m$  are the snow and melting temperature of snow, respectively. We calculate LW<sub>bc</sub> using Eq. (2):

$$LWbc = (CC/Lf) - (SWE(sw) - LWC),$$
(2)

where CC is cold content (MJ/m²), SWE is snow water equivalent (kg/m²),  $S_w$  is the irreducible water saturation of the snowpack (percent by weight), LWC is liquid water content of the snowpack (kg/m²), and  $L_f$  is the latent heat of fusion for water (0.334 MJ/kg). Note that the units of  $LW_{bc}$  (kg/m²) can be expressed as millimeters of water equivalent (mm WE), which we use for our results.

Laboratory derived estimates of  $S_w$  range from 5-15% by weight (e.g. Coléou and Lesaffre, 1998). We identify  $S_w$  in the model output as this is the minimum liquid water content within a zero-degree snowpack that allows for runoff. We calculate this for each grid point by determining the annual minimum of the liquid water content divided by SWE where (1) SWE is greater than zero, (2) cold content is zero, and (3) water flows from the base of the snowpack. This

value of  $S_w$  allows for spatial and temporal variation of irreducible water saturation without the magnitude of LW<sub>bc</sub> falling below zero.

## 2.4 Trend analysis

We examine trends in modeled SWE, cold content, and  $LW_{bc}$  over the 72 years of model time (water years 1951-2022). Each of these snowpack parameters varies throughout the year at each grid point, generally growing slowly in magnitude from late fall through mid-winter and reducing quickly in the late winter and spring. Since this evolution of accumulation and ablation is roughly consistent through time, we are able to analyze inter-annual variations in snow properties by date. We exclude Feb 29<sup>th</sup> from our analyses which is consistent with previous snow studies that assess seasonal snowpack on a specific date, usually early to mid-April in the Western United States (e.g. Bohr and Aguado, 2001; Hatchett and McEvoy, 2018; Pederson et al., 2011; Schmitt et al., 2024; Serreze et al., 1999).

Snowpack properties on any given day can vary greatly from year to year (e.g. Fig. 2a), resulting in low significance to a trend line fit to data. Therefore, to generate robust confidence intervals in our trend analysis, we use a modified bootstrap linear regression technique wherein we randomly omit a portion (20%) of the data, compute a linear fit to the remaining data, and repeat the process 10,000 times for each day of the water year. There are approximately 20 billion random combinations of a 20/80 split for our time series data; we assume that 10,000 iterations are a representative sub-set of the full set of random combinations (e.g. Fig. 2a). Histograms of regression coefficients (slopes) for the linear fits reveal a near-Gaussian probability distribution of trends associated with the change in modeled snowpack properties on a single day of year (e.g. Fig. 2b). Trends and errors are then based on the evolution of the daily probability distributions throughout the water year.

Figure 2: Example of a single day (April 1) trend analysis of  $LW_{bc}$ . (a) Full-basin mean  $LW_{bc}$  values for each year (red plus signs) with the regression lines (aqua with 1% opacity) from fitting a random 80% of the data 10,000 times. (b) A histogram of the slopes of 10,000 regression lines plotted in (a) reveals a near-Gaussian (orange dotted line) distribution.

This modified bootstrap approach allows us to constrain the upper and lower significance bounds for the full 72-year period more accurately than other slope estimation techniques such as the Theil-Sen estimator, which includes short time scales in the error calculation. Both methods give similar values for coefficient of change (less than 0.25 mm WE difference when applied to SWE data). Here we consider trends to be statistically significant only where the 2σ, 95% bounds on the normal distribution of the regression coefficients do not include 0 mm WE per year.

155

Along with providing an analysis of the 72-year trend in average snowpack over the entire basin, we also analyzed the trends over 17 elevation bands of 100 m each to elucidate the effect of elevation on the snowpack properties. To aid in our analysis of the trends, we divide the snow season into four periods which are defined by inflection points in the SWE and LW<sub>bc</sub> trend curves (see results). We call these periods Early Accumulation (November 19 to January 27), Core Accumulation (January 28 to March 8), Late Accumulation (March 9 to April 17), and Melt Onset (April 18 to June 7).

# 3 Results

# 3.1 Seasonal evolution of trends

Snowpack properties from our model output show interannual variability greater than 60% of mean basin values for each day from the beginning of Early Accumulation to through the end of Melt Onset (e.g. Fig. 2a). However, even with the large variance in the data, probability distribution functions of the modified bootstrap linear fits are robust, resulting in a near-Gaussian distribution of the calculated trends (e.g. Fig. 2b). The prevalence of large interannual variability resulting in near-normal distribution from Early Accumulation through Melt Onset is seen in SWE, CC, and  $LW_{bc}$  (Fig. 3). For modeled SWE outputs, these variations in magnitude are approximately commensurate with the mean magnitude of values whereas variations in magnitude are greater than the mean magnitude of CC and  $LW_{bc}$  daily values.

Trends in snowpack properties exhibit distinct seasonal patterns. Across the Early Accumulation Period, the magnitudes of SWE trends are negative, indicating diminishing snowpack over the 72 years. With changes amassing as the snowpack accumulates during winter, the trend magnitudes of SWE loss generally grows with time, reaching a rate of  $1.4 \pm 0.5$  mm WE yr<sup>-1</sup> by the end of the period. The SWE loss is accompanied by a small reduction in CC of up to  $0.050 \pm 0.02$  mm WE yr<sup>-1</sup>, and small but significant trends of decreasing LW<sub>bc</sub>.

Figure 3: Model results for SWE (a), CC (c), and  $LW_{bc}$  (e) for the entire SFF basin. In each panel, the lighter lines show yearly evolution of the individual snowpack property, black lines are the 72-year mean values. Daily trend probability values for SWE (b), CC (d), and  $LW_{bc}$  (f) are plotted with mean values (darkest lines), 90% confidence intervals (darker shading), and 95% confidence intervals (light shading). These probability values are derived from the gaussian distribution of trends like those shown in Fig. 2. Both CC and  $LW_{bc}$  are values of energy deficiency that must be overcome before snowpack runoff can occur, therefore reported values are negative and positive values in the coefficient of change (trend) indicate a reduction of the magnitude of the property. For visual consistency with changes in SWE, we invert the y-axis of trend plots for CC and  $LW_{bc}$  so that values below the x-axis indicate a decrease in absolute magnitude.

The Core Accumulation Period begins with held-over deficits in SWE, CC, and  $LW_{bc}$ . Trend magnitudes of SWE loss cease to increase over time, indicating no significant changes to the growth of SWE during the midwinter period. However, CC trends, which start the period as negative, become increasingly less negative early in the period. This indicates that the CC during the 72 years trended upward during much of February due to colder snowpack temperatures. The magnitude of  $LW_{bc}$  trends show slight decreases during early February but remain relatively unchanged during the mid-winter period. The trends of CC loss then increase in the Late Accumulation Period, likely reflecting greater surface temperatures and/or more liquid precipitation. As a result,  $LW_{bc}$  trend magnitudes decline sharply despite unchanged trends in SWE.

195

During the Melt Onset Period, SWE trends show rapidly decreasing values. This is consistent with earlier melt onset; the date of peak SWE shifted by 0.18 days earlier per year, equivalent to about two weeks earlier over 72 years (Fig. A2). The trends of CC approach zero by late May, reflecting no possible changes in the CC of an isothermal snowpack. Similarly, trends of LW<sub>bc</sub> continues to decline as the snowpack ripens and becomes isothermal.

# 3.2 Elevation analysis

We find increased vulnerability to ROS flooding events at all elevations during all accumulation periods (Fig. 4a), with the largest trends occurring at the highest elevations (Fig. 4b) and the greatest risk of a total loss of  $LW_{bc}$  at relatively low elevations since the  $LW_{bc}$  magnitude is less at low elevations (Fig. 4c). Percent loss of  $LW_{bc}$  in the Early and Core Accumulation periods is approximately 20% of the mean magnitude of  $LW_{bc}$  with the exception of the lowest elevation band. Late Accumulation percent loss of  $LW_{bc}$  ranges from about 42% to 78%, with the highest percent loss at the lowest elevations (Fig. 4a). Increased risk of ROS-induced runoff causing flooding, however, is greatest where the percent decrease in  $LW_{bc}$  is high and the mean magnitude SWE is also high. Our data show that this elevation range is between about 1600 m and 2300 m, where modeled  $LW_{bc}$  has decreased by 42% to 58% and over 75% of total basin SWE typically exists (Fig. 4d).

Figure 4: Total change in  $LH_{bc}$  over the 72-year period as a percentage of mean magnitude at 100 m elevation bands (a) for Early Accumulation (blue circles, dashed line), Core Accumulation (orange X's, dotted line), and Late Accumulation (green triangles, solid line). These values were calculated by multiplying the mean trend (b) by 72 years and dividing the result by the mean magnitude (c) for each elevation bin. Although the lowest elevations have the highest percent change in  $LW_{bc}$ , very little of the basin SWE is located in the lower elevations (d), meaning a ROS melt event will have minimal impact compared to ROS melt events over elevation rages with higher percent of total basin SWE.

# 3.3 Long-term changes in snowpack buffering

220

Examination of mean LW<sub>bc</sub> over the 72 model years reveals an increasing amount of interannual variability for mean LW<sub>bc</sub> during the Late Accumulation period which is not apparent in the Early or Core Accumulation periods (Fig. 5 a,b,c). Changes in interannual variability are especially striking when we compare the coefficient of variation (CV) for the first 36 model years (1950-1986) to the last 36 model years (1987-2022) over the three accumulation periods (Table 1). This comparison reveals that there is very little change in LW<sub>bc</sub> interannual variability during the Early Accumulation and Core Accumulation periods, whereas the last 36 years of the Late Accumulation period has nearly double the interannual variability (52.1%) of the first 36 years (27.1%).

Figure 5: Evolution of mean annual  $LW_{bc}$  for Early Accumulation (a), Core Accumulation (b), and Late Accumulation (c). Note the significant decrease in minimum Late Accumulation mean  $LW_{bc}$  from 1986 (where Late Accumulation  $LW_{bc}$  is zero) to 2022. In fact, 16 of the 36 years after 1986 have lower mean Late Accumulation  $LW_{bc}$  than any year before 1986. This level of decrease is not seen in the fall and Core Accumulation data. (d) Plotting the number of years with at least one Early Accumulation (March 10 to April 18) day with zero  $LW_{bc}$  at each elevation reveals the discrepancy between the first 36 years (water years 1951-1986) and the second 36 years (1987-2022). This discrepancy is evident at all elevations.

Table 1: Coefficient of variability for the first 36 and last 36 model years in the Early Accumulation, Core Accumulation, and Late Accumulation periods.

| Period             | WY 1951-1986 | WY 1987-2022 |
|--------------------|--------------|--------------|
| Early Accumulation | CV = 30.7    | CV = 28.1    |
| Core Accumulation  | CV = 23.7    | CV = 26.4    |
| Late Accumulation  | CV = 27.1    | CV = 52.1    |

# 3.4 Differences in 1950-1985 and 1986-2022

There are other marked differences between the Late Accumulation snowpack  $LW_{bc}$  for the first 36 model years and last 36 model years. To highlight this, we examined the number of years with at least one day of zero  $LW_{bc}$  for each 100 m elevation band within the basin (Fig. 5d). During the first 36 model years, at the lowest elevations (900 m to

240 1300 m) Late Accumulation LW<sub>bc</sub> reached zero for at least one day between 25% and 97% of modeled years, between 1300 m and 2000 m spring LW<sub>bc</sub> reached zero between 3% and 9% of the years. Only the year 1986 had zero LW<sub>bc</sub> for at least one day above 2000 m elevation, the 1986 model run has zero LW<sub>bc</sub> for the entire Late Accumulation period across all elevations. In contrast, the last 36 years of the model period has 2-5 years where LW<sub>bc</sub> went to zero for at least one day at all elevations. Further, below 1300 m elevation LW<sub>bc</sub> dropped to zero more than 50% of years, between 1300 and 2100 m elevation 14% to 44% of the latter half of the years modeled had zero LW<sub>bc</sub>, and 5% to 11% of years had zero LW<sub>bc</sub> for at least one day for elevations above 2100 m.

#### 4 Discussion

#### 4.1 Infiltration processes

Interpretation of the trends we identify in our analysis must accommodate the limitations of our modeling framework and assumptions. Our analysis addresses the total  $LW_{bc}$  of the full one-dimensional snowpack, calculated at points and distributed over the basin. However, prior studies have shown that during ROS events, initial water inputs to the snowpack often bypasses much of the snow by traveling along preferential flow paths, reaching the base and initiating runoff before the entire snowpack is saturated or "ripened" (Juras et al., 2017; Würzer et al., 2016). Runoff can therefore begin several hours before the bulk snowpack has received enough energy to become fully ripened. In our case, the 1D  $LW_{bc}$  can be interpreted either as the energy barrier associated with flow along preferential pathways, or as the integrated capacity of the full snowpack. The distinction is critical in hydrologic studies concerning the buffering effects on the specific timing of ROS runoff onset.

# 4.2 Snowpack trends

Trends of declining peak SWE in the mountain snowpack of the western U.S. have been documented by numerous studies (e.g. Hamlet et al., 2005; Mote et al., 2005, 2018), most commonly based on April 1 values. However,  $LW_{bc}$  and trends therein depend on the interplay between SWE and CC and thus require more complex interpretation with regards to changes in potential runoff from ROS events.

High interannual variability is SWE is a well-established characteristic of the Western U.S. mountain snowpack (e.g. Cayan, 1995; Musselman et al., 2021; Serreze et al., 1999; Zeng et al., 2018). Our results show an overall decline in LW<sub>bc</sub> that is accompanied by increased interannual variability, especially in spring. Some years are characterized by shallow, warm snowpacks, while others resemble the colder, more persistent snowpacks that were common in the 1950s–1980s. Thus, despite a long-term trend toward greater ROS-induced runoff risk due to lowered LW<sub>bc</sub>, individual years with relatively low susceptibility still occur late in the record. Overall trends need to be interpreted within the context of year-to-year variability.

The impact of changing  $LW_{bc}$  on the potential for ROS-induced runoff also shows strong seasonality. Our findings reveal a clear trend toward less snowpack during the Early Accumulation period, and thus reduced average flood

275

hazard, consistent with prior work indicating that ROS risk declines with diminishing snowpack (McCabe et al., 2007; Musselman et al., 2018). Although the effects of reduced SWE and CC during the Early Accumulation period carry over to the Core Accumulation period, the latter shows no additional loss in LW<sub>bc</sub> over our 72-year record. SWE exhibits no significant trend, and CC actually increases over much of the period, as does LW<sub>bc</sub> early on. Had significant warming or rainfall increases occurred, we would expect declining CC and LWbc, which are not observed. This stability is remarkable compared to the substantial changes in snowpack properties during the early and late periods. Moreover, despite representing a relatively small share of the snowpack's energy balance, the CC appears to play a critical role in maintaining this resilience.

Although SWE in our basin peaks in later April or May, typical of the Northern U.S. Rockies (e.g. Bohr and Aguado, 2001; Zanewich and Rood, 2025), March emerged in our analysis as the critical month with relatively sustained peak LW<sub>bc</sub>. After March, the average potential for runoff from ROS events has increased as the CC has trended strongly downward and the number of days with a ripe snowpack has trended upward. While the largest changes in LW<sub>bc</sub> were at high elevations, this is where the LW<sub>bc</sub> is greatest and requires the most energy input from a ROS event to induce runoff. Low elevation areas of the basin, on the other hand, do not accumulate much SWE and never had much LW<sub>bc</sub> against ROS flooding. Thus, it is the mid-to-high elevations that represent a "vulnerability zone," where significant SWE volume, combined with marked LW<sub>bc</sub> decline, creates conditions most conducive to ROS enhanced runoff. These areas contribute the bulk of basin SWE, making their increased susceptibility particularly important for downstream flood risk.

## 4.3. Trends compared to ROS event size

Quantifying the historical frequency, duration, and intensity of winter rainfall events in mountain basins remains challenging, particularly in basins lacking long-term instrumentation. While extreme weather events are already intensifying (e.g. Fischer and Knutti, 2016; Li et al., 2024) and are projected to further increase under a warming climate (Seneviratne et al., 2021; de Vries et al., 2024), the rarity and spatial variability of such events make them particularly difficult to quantify and project at the scale required for watershed hazard assessments (e.g. Broderick et al., 2019). Conceptually, a warming climate tends to (a) shift precipitation from snow to rain and (b) increase the magnitude of extreme precipitation events (Donat et al., 2013; Rahmstorf and Coumou, 2011). However, the detailed quantification of these metrics for a given basin, which is necessary for robust rain-on-snow (ROS) risk analysis, remains elusive.

Nevertheless, the relevance of our findings depends on whether the observed 72-year decline in snowpack  $LW_{BC}$  is meaningful relative to the size and frequency of winter rainfall events in this basin. This motivates a first-order assessment of snowpack  $LW_{BC}$  with precipitation. Over the study period, the March 15 snowpack  $LW_{BC}$  declined on the average trend line by 19 mm, from 46 mm in 1950 to 27 mm in 2022. Thus, a rainfall event of 27 mm would deplete all the 2022 snowpack  $LW_{BC}$  value, but only 60% of the 1950 value which requires 46 mm. How frequent have such events been? Using the 49 years of existing precipitation records from the Noisy Basin SNOTEL site, March

events with above-freezing air temperatures had recurrence intervals of approximately 2 years for 27 mm and 10 years for 46 mm. Many of these events were likely warm snowfalls rather than pure rain. Nevertheless, the decline in snowpack LW<sub>BC</sub> along the trend line implies an impactful increase (fivefold in this exercise) in the frequency that the snowpack LW<sub>bc</sub> could be fully depleted by a precipitation event. Importantly, this appraisal (1) is based on changes to the snowpack only, not precipitation; (2) assumes that all precipitation falls as rain, and; (3) applies only to the mean trend, noting that snowpack LW<sub>BC</sub> exhibits substantial interannual variability.

# 5. Conclusions

Using distributed, physics-based modeling of snowpack evolution driven by 72 years of climate reanalysis data, we evaluate long term changes in snowpack properties that serve to buffer ROS water inputs from runoff. Our trend analysis, via a modified bootstrap regression, reveals overall declining LW<sub>bc</sub> in a system characterized by very high interannual variability. This trend signals an increased susceptibility of the average winter snowpack to runoff from ROS events. Snowpack changes, however, have not been monotonic with regards to the winter season: declines in buffering capacity are concentrated in fall and spring, while mid-winter conditions have remained comparatively unchanging.

Trends showed largest losses of cold content and LW<sub>bc</sub> during the Late Accumulation and Melt Onset periods. Consequently, the number of days with ripe snowpack and elevated ROS risk increased substantially. On average, late-season LW<sub>bc</sub> decreased by 42–80% relative to the 72-year mean, with lower elevations experiencing the largest proportional declines. The interannual variability of LW<sub>bc</sub> during the Late Accumulation period intensified in recent decades, with some years resembling deep and cold snowpacks of the 1950–1980s and others showing exceptionally low buffering capacity.

Our basin study serves as a proof of concept, showing changes in snowpack properties can be strongly sub-seasonal and elevation dependent. This suggests that assessments of ROS risk are not easily generalized by region (e.g., coastal versus intermountain) but must explicitly account for both the timing within the snow season, the controlling role of elevation, and the importance of interannual variability. High fidelity assessments of snowpack physical conditions are thus needed to meaningfully anticipate evolving flood hazards in a warming climate.

# **Author contribution**

JH conceived the original idea for the project. JB performed modelling, developed and employed trend analysis, and created all figures and tables. JB and JH contributed to interpretation of the results and wrote the manuscript.

# Code and Data Availability

Code written for this study is available at: https://doi.org/10.5281/zenodo.17290309. ERA5-Land data can be downloaded at: https://cds.climate.copernicus.eu/datasets/reanalysis-era5-land?tab=download. Alpine3D code is available at: https://code.wsl.ch/snow-models/alpine3d.

# 345 Acknowledgements

This work was funded by NSF grant number 2119689

# **Competing Interests**

The authors have no competing interests.

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

# Appendix A

Table A1: List of required input variables for Alpine3D, the source of the variable, and the equation used to calculate values from the source.

| Alpine3D input variable       | Variable name(s)                        | Variable Symbol(s) | Source    | Equation                                                                          |
|-------------------------------|-----------------------------------------|--------------------|-----------|-----------------------------------------------------------------------------------|
| Timestamp                     | Timestamp                               | Т                  | ERA5-Land | _                                                                                 |
| Air temperature               | 2m temperature                          | $t_a$              | ERA5-Land | _                                                                                 |
| Relative humidity             | 2m dewpoint temperature, 2m temperature | $t_d, t_a$         | ERA5-Land | $100*\frac{e^{\frac{17.625*t_d}{243.04*t_d}}}{e^{\frac{17.625*t_a}{243.04*t_a}}}$ |
| Ground temperature            | Soil temperature level 1                | $t_s$              | ERA5-Land | _                                                                                 |
| Wind direction                | 10 m u-component of wind, 10            | u, v               | ERA5-Land | $\arctan \frac{u}{v} - \frac{\pi}{2}$                                             |
|                               | m v-component of wind                   |                    |           |                                                                                   |
| Wind speed                    | 10 m u-component of wind, 10            | u, v               | ERA5-Land | $\sqrt{v^2 * u^2}$                                                                |
|                               | m v-component of wind                   |                    |           |                                                                                   |
| Total precipitation           | Total precipitation                     | $P_{tot}$          | ERA5-Land | $P_{tot} * 1.5$                                                                   |
| Incoming short-wave radiation | Surface solar radiation down-           | SSRD               | ERA5-Land | SSRD*3600                                                                         |
|                               | wards                                   |                    |           |                                                                                   |
| Incoming long-wave radiation  | Surface thermal radiation               | STRD               | ERA5-Land | STRD*3600                                                                         |
|                               | downwards                               |                    |           |                                                                                   |
| Elevation                     | Geopotential                            | Geopotential       | ERA5      | Geopotential/9.80665                                                              |
| X-Location                    | Latitude and Longitude                  | Lat, Lon           | ERA5-Land | WGS84 to UMT12N trans-                                                            |
|                               |                                         |                    |           | form                                                                              |
| Y-Location                    | Longitude and Longitude                 | Lat, Lon           | ERA5-Land | WGS84 to UTM12N trans-                                                            |
|                               |                                         |                    |           | form                                                                              |
|                               |                                         |                    |           |                                                                                   |

Relative humidity equation constants are from Alduchov and Eskridge (1996), dewpoint and air temperatures are in degress Celsius. Radiation values from ERA5-Land data are given as average  $\frac{W/m^2}{s}$ , Alpine3D uses total radiation over each timestep, thus we multiply the average radiation data by 3600  $\frac{s}{hour}$ . Location transforms were applied in QGIS.

Figure A1 – Comparison of Alpine3D model output and SNOTEL maximum SWE (a) and day of maximum SWE (b) for water years 1981-2023. Here we show the low elevation Emery Creek SNOTEL (orange triangles) and higher elevation Noisy Basin SNOTEL (blue circles) compared to the model output grid point within 1 km of the SNOTEL sites that are the closest to the SNOTEL elevations. ERA5-Land precipitation values were multiplied by 1.5 for input to Alpine 3D to achieve this level of fit.

Figure A2 – Modified bootstrap analysis showing a shift in the date of maximum SWE (a) with the distribution of trends (b). Day of water year (a) is number of days since October 1. The Gaussian distribution for the data in (b) is shown as a dashed orange line.

2010

510

1950 1960 1970