# Peer review of "Historical Evolution of Snowpack Capacity to Buffer Rain-on-Snow Runoff in a Large Columbia River Headwaters Basin"

_EGUsphere, 2025_

## Referee Comment (RC2)

Overall, this is an interesting and novel study that investigates the impact of snowpack dynamics on liquid water buffering using a physics based model.

Overall, I had some major concerns with the written manuscript, as a lot of information seems to be missing that rendered it difficult for the reader to easily understand the results. I am recommending a major revision.

My main concerns are: Some additional justification and background is needed in the introduction, there needs to be a data section in the methods, model evaluation/accuracy using RMSE, MAE, and/or bias needs included and the process explained in a new section,

See my overall comments below:

**Abstract:**

Line 12 – I recommend including the model name instead of a "a snowpack model"

Line 16: There needs to be a brief explanation why there is a long-term decline in LWbc.

**Introduction**

Line 61: I am recommending a paragraph introducing and justifying the model used in this research. It should include literature supporting that it is the most appropriate model for this research question and justify the choice.

~Line 61- Another paragraph needs added that provides background on SNOTEL including an explanation of SNOTEL and the data that are extracted from it and justifies with literature ERA5 as input to the model.

Line 65 (e.g. final paragraph of introduction): The objections paragraph here should also include that model – perhaps after/with this sentence? "72-year model simulation of the seasonal evolution of internal snowpack conditions"

**Methods:**

Line 94: something seems to be missing here: "a spatially distributed snow allocation and surface hydrology …. Model?"

Line 90 New Section 2.2: Data - There needs to be a data section where you explain the SNOTEL and ERA5 data. You also need to include an explanation of "beginning of Early Accumulation to through the end of Melt Onset". Also an explanation of early, core, and late accumulation – what specifically was used to define these?

Line 98: ERA5 needs its own paragraph

Section 2.2: Snowpack Simulations – I'm a little unclear here – in the trends, you mention that you have modelled SWE, cold content, and LWbc – however, this is not described in this section. Therefore, I am requesting the following:

1. You need to include an explanation of the input parameters, the input data, and an explanation of the variables that were modelled, and an explanation of the outputs.

Line 126: A new Section 2.3: A new section needs added here for model error. I am requesting the following:

1. Please add an RMSE, MAE, and bias between your simulated data and the SNOTEL data to quantify model accuracy. You need to ensure that you simulate a variable that is also available within the SNOTEL data so that you can quantify model accuracy
2. Add a new section before trend analysis that describes your process for model accuracy

**Results:**

Line 165: "beginning of Early Accumulation to through the end of Melt Onset" were not defined in the methods. Are these specific days that you chose for the analysis?

Line173: Need to include figure reference – I recommend including individual figure references (e.g. 3a, 3b, 3c) because its hard to cross reference currently.

Line 201: I like how you reference figure 4a, 4b, 4c, etc. all separately in this section and I am recommending a similar format in section 3.1

Section 3.4: This seems like it could use a figure.

**Discussion:**

I thought this was written rather well.